# Implementation of Microcirculation Examination in Clinical Practice—Insights from the Nationwide POL-MKW Registry

**DOI:** 10.3390/medicina60020277

**Published:** 2024-02-05

**Authors:** Rafał Januszek, Łukasz Kołtowski, Mariusz Tomaniak, Wojciech Wańha, Wojciech Wojakowski, Marek Grygier, Wojciech Siłka, Grzegorz Jan Horszczaruk, Bartosz Czarniak, Radosław Kręcki, Bartłomiej Guzik, Jacek Legutko, Tomasz Pawłowski, Paweł Wnęk, Marek Roik, Sylwia Sławek-Szmyt, Miłosz Jaguszewski, Tomasz Roleder, Miłosz Dziarmaga, Stanisław Bartuś

**Affiliations:** 1Faculty of Medicine and Health Sciences, Andrzej Frycz Modrzewski Cracow University, 30-705 Kraków, Poland; 21st Department of Cardiology, Medical University of Warsaw, 02-091 Warsaw, Poland; lukasz@koltowski.com (Ł.K.); tomaniak.mariusz@gmail.com (M.T.); 3Department of Cardiology and Structural Heart Diseases, Medical University of Silesia, 40-055 Katowice, Poland; wojciech.wanha@gmail.com (W.W.); wojtek.wojakowski@gmail.com (W.W.); 41st Department of Cardiology, Poznan University of Medical Sciences, 61-701 Poznań, Poland; mgrygier@wp.pl (M.G.); sylwia.slawek@usk.poznan.pl (S.S.-S.); 5Faculty of Medicine, Jagiellonian University Medical College, 31-008 Kraków, Poland; wojciech.silka@student.uj.edu.pl (W.S.); stanislaw.bartus@uj.edu.pl (S.B.); 6Faculty of Medical Science, Collegium Medicum. Cardinal Stefan Wyszyński University in Warsaw, 01-938 Warsaw, Poland; horhor@wp.pl; 7Department of Cardiology, Voivodeship Hospital in Łomża, 18-404 Łomża, Poland; 8Provincial Specialist Hospital in Wloclawek, 87-800 Włocławek, Poland; bartoszczarniak@gmail.com; 9Department of Cardiology, Scanmed, 99-320 Kutno, Poland; rkrecki@ptkardio.pl; 10Department of Interventional Cardiology, Institute of Cardiology, Jagiellonian University Medical College, św. Anny 12, 31-007 Kraków, Poland; b.guzik@uj.edu.pl (B.G.); jacek.legutko@uj.edu.pl (J.L.); 11Department of Interventional Cardiology, The John Paul II Hospital, Prądnicka 80, 31-202 Kraków, Poland; 12Department of Cardiology, National Institute of Medicine of the Ministry of Internal Affairs and Administration, 02-507 Warsaw, Poland; pawtom@gmail.com; 13Centre of Postgraduate Medical Education, 01-813 Warsaw, Poland; 14Provincial Specialist Hospital in Wroclaw, 51-124 Wrocław, Poland; pawel.wnek1@gmail.com; 15Department of Internal Medicine and Cardiology, Medical University of Warsaw, 02-091 Warsaw, Poland; mroik@wum.edu.pl; 161st Department of Cardiology, Medical University of Gdańsk, 80-210 Gdańsk, Poland; mjaguszewski@gumed.edu.pl; 17Department of Cardiology, Wroclaw Medical University, 50-556 Wrocław, Poland; tomaszroleder@gmail.com; 18Department of Cardiology-Intensive Therapy and Internal Diseases, Poznan University of Medical Sciences, 60-355 Poznań, Poland; milosz.dziarmaga@usk.poznan.pl

**Keywords:** coronary flow reserve, coronary microvascular dysfunction, index of microcirculatory resistance, microcirculation

## Abstract

*Background and Objectives:* The assessment of coronary microcirculation may facilitate risk stratification and treatment adjustment. The aim of this study was to evaluate patients’ clinical presentation and treatment following coronary microcirculation assessment, as well as factors associated with an abnormal coronary flow reserve (CFR) and index of microcirculatory resistance (IMR) values. *Materials and Results:* This retrospective analysis included 223 patients gathered from the national registry of invasive coronary microvascular testing collected between 2018 and 2023. *Results:* The frequency of coronary microcirculatory assessments in Poland has steadily increased since 2018. Patients with impaired IMR (≥25) were less burdened with comorbidities. Patients with normal IMR underwent revascularisation attempts more frequently (11.9% vs. 29.8%, *p* = 0.003). After microcirculation testing, calcium channel blockers (CCBs) and angiotensin-converting enzyme inhibitors were added more often for patients with IMR and CFR abnormalities, respectively, as compared to control groups. Moreover, patients with coronary microvascular dysfunction (CMD, defined as CFR and/or IMR abnormality), regardless of treatment choice following microcirculation assessment, were provided with trimetazidine (23.2%) and dihydropyridine CCBs (26.4%) more frequently than those without CMD who were treated conservatively (6.8%) and by revascularisation (4.2% with *p* = 0.002 and 0% with *p* < 0.001, respectively). Multivariable analysis revealed no association between angina symptoms and IMR or CFR impairment. *Conclusions:* The frequency of coronary microcirculatory assessments in Poland has steadily increased. Angina symptoms were not associated with either IMR or CFR impairment. After microcirculation assessment, patients with impaired microcirculation, expressed as either low CFR, high IMR or both, received additional pharmacotherapy treatment more often.

## 1. Introduction

Epicardial arteries larger than 400 µm in diameter constitute approximately 5% of coronary macro- and microcirculation; the remaining 95% is made up of pre-arterioles, arterioles and capillaries. The term microvascular angina was first introduced into the nomenclature in 1985 by Cannon and Epstein [1]. Structural and functional abnormality of the microvasculature system, also referred to as coronary microvascular dysfunction (CMD), is associated with multiple diseases, as well as infarct size, and overall poor clinical outcomes in many patient subgroups, e.g., those presented with myocardial infarction (MI) [2,3,4,5,6,7,8,9]. However, the largest group of patients are those with chronic coronary syndromes, persistent symptoms and clinical signs of heart ischemia, where there is no evidence of obstructive coronary artery disease based on angiography (INOCA). It is estimated that up to 40% of patients undergoing diagnostic coronary angiography fall into this group of patients [10]. Among women, this may constitute up to 2/3 of patients who have typical ischaemia symptoms without atherosclerotic lesions visible in coronary angiography [11]. Furthermore, it is estimated that up to 1/13 of these people will die within 10 years of the angiography, with the most common cause of hospitalisation being heart failure [12,13]. Additionally, approx. 70–80% of patients with chest pain and no obstructive coronary artery disease (CAD) present evidence of diffuse non-obstructive atherosclerosis, as well as coronary calcifications, in intravascular imaging [14,15]. In 2007, Camici and Crea proposed clinical and pathogenetic classifications of CMD, which, based on a clinical setting, distinguished the following four types of CMD: that which occurs in the absence of myocardial diseases and obstructive CAD, that which occurs in the presence of myocardial diseases, that which occurs in obstructive CAD, and those which are related to iatrogenic etiology [16].

As of now, two primary measures are prevalent in functional microvasculature testing: the coronary flow reserve (CFR) and the index of microcirculatory resistance (IMR). While CFR provides insight into the functions of both epicardial coronary vessels and distal vasculature, IMR offers direct assessment of the microcirculation function. Both measures are recommended by international guidelines as diagnostic methods for identifying patients with CMD, especially in the case of non-obstructive CAD, as confirmed by the normal fractional flow reserve (FFR) [17,18,19]. Cut-off values of 2.0 for CRF and 25 for IMR have been widely adopted [20,21].

In many studies, the importance of routinely assessing microcirculation is emphasised, as it may provide additional insight into a patient’s disease, and can be of additional prognostic value [9,20,22]. However, there are limited data regarding the characteristics of patients who undergo microcirculation testing.

In our study, we aimed to compare clinical characteristics and subsequent decision-making after assessing coronary microvasculature between patients with abnormal microcirculation function and those with normal IMR and CFR values. Moreover, we sought to evaluate factors associated with impaired microvasculature measurements.

## 2. Materials and Methods

This retrospective study is based on a national registry of invasive coronary microvascular testing (POL-MKW), in which 223 patients have been gathered to date. Data have been collected from eight catheterisation laboratories (CathLabs) in Poland between 2018 and 2023.

### 2.1. Coronary Angiography and Physiological Examination of Coronary Arteries

The access site, sheath and catheter size, as well as periprocedural anticoagulation use, were per operator preference. Measurements were carried out using a dedicated pressure guide (Abbott PressureWire™ X Guidewire) and the Coroventis CoroFlow Cardiovascular System. Cardiovascular medications were discontinued for 24 h prior to microcirculation assessment. A therapeutic dose of unfractionated heparin was administrated, i.e., 5000 units or 80–100 units per kg during the intervention.

Fractional Flow Reserve (FFR) was calculated as the lowest average Pd/Pa from three consecutive heartbeats during maximal hyperaemia. CFR was calculated as the ratio of mean transit time (Tmn) at rest/hyperaemic Tmn. IMR was calculated from the Pd × Tmn equation determined during hyperaemia. All the calculations mentioned above were carried out automatically by the software. IMR values ≥ 25 and CFR ≤ 2 were adopted as abnormal and defined as CMD. Patients with acute myocardial infarction (at least 72 h after diagnosis and depending on the extent of ischaemia) were excluded from the study. In cases where collateral flow should be taken into account (presence of tight stenosis in the assessed vessel or collaterals to the vessel with chronic total occlusion)—IMR was corrected according to formulas proposed by Yong et al. [23].

### 2.2. Statistical Analysis

Nominal variables are presented as absolute numbers and percentages. Continuous variables are expressed as means (standard deviation) and medians [first quartile; third quartile], depending on their normality, which was evaluated using the Shapiro–Wilk test. For normally distributed, continuous variables, differences were compared via the Student’s or Welch’s *t*-tests. In the case of non-parametrical data, the Wilcoxon test was used instead. Categorical variables were compared using Pearson’s chi-squared or Fisher’s exact test if 20% of the cells had an expected count of less than five (Monte Carlo simulation for Fisher’s test using tables of higher dimensions than 2 × 2).

All factors that may have been associated with abnormal IMR and CFR values were adopted in univariable logistic regression models. Based on their results, variables with a *p*-value < 0.2 were subsequently included in the multivariable model, having risk estimates presented as odds ratios (OR) with 95% confidence intervals (CI). A *p*-value lower than 0.05 was considered significant. The entire statistical analysis was carried out using the R test, version 4.3.1 (R Core Team (2023). R: a language and environment for statistical computing. R Foundation for Statistical Computing, Vienna, Austria. URL https://www.R-project.org/ (accessed on: 11 July 2023)).

## 3. Results

### 3.1. Catheterisation Laboratories and Years of Study

As presented in Figure 1, the frequency of performed coronary microcirculatory assessments using the Coroventis system has increased since 2018, reaching its peak in 2022 (Figure 1). These data are from July 2023; therefore, more assessments are anticipated to be conducted in 2023. Furthermore, the vast majority of assessments were performed at the four dominant centres located in Warsaw, Poznań and Kraków (Figure 2).

### 3.2. General Characteristics and Concomitant Diseases

We studied 223 patients with a median age of 66.2 [59.9; 71.9], most of whom were males (55%) (Appendix A). Patients with abnormal IMR were less likely to have experienced a prior MI (19.3% vs. 42.1%, *p* < 0.001), prior percutaneous coronary intervention (PCI) (17.9% vs. 43.7%, *p* < 0.001), and were less often burdened with ischaemic heart disease (53.6% vs. 89.9%, *p* < 0.001) or thyroid problems (*p* = 0.04), as compared to the group with normal IMR. However, atrial fibrillation (27.4% vs. 14.9%, *p* = 0.03) and prior pulmonary embolism/deep venous thrombosis (PE/DVT) (13.1% vs. 3.4%, *p* = 0.01) were more prevalent in these patients, with the latter also being confirmed in the group with low coronary flow (12.2% vs. 3.5%, *p* = 0.02, Appendix A).

### 3.3. Clinical State and Main Symptoms

Angina-related symptoms, including shortness of breath, chest pain and palpitations, were significantly more prevalent in the group with either IMR or CFR impairment (Table 1). The distribution of the CCS class differed between patients having normal and impaired IMR, with the latter group demonstrating severe symptoms (Table 1). More detailed data regarding findings from examinations, e.g., echocardiography, are shown in Appendix A.

### 3.4. Pharmacotherapy

As compared to the group with normal IMR, patients with abnormal IMR had less often taken P2Y12 inhibitors *(p* = 0.002) or calcium channel blockers (23.2% vs. 39.2%, *p* = 0.02, Appendix A).

### 3.5. Coronary Angiography

The group with IMR abnormality was characterised by a lower prevalence of intermediate stenosis (30–70%) (41% vs. 73.7%, *p* < 0.001) and chronic total occlusions (1.2% vs. 9.1%, *p* = 0.02), as compared to the group with normal IMR. More detailed data are shown in Table 2.

### 3.6. Coronary Microvascular Circulation Assessment

In the total population, the median CFR was 2.3 [1.6; 3.4], while the median IMR was 20.0 [13.0; 33.0] (Appendix A). Patients with abnormal IMR had higher median values of resting full-cycle ratio (RFR) and FFR compared to the group with normal IMR (*p* < 0.001 and *p* = 0.006, Appendix A). Moreover, lower CFR and higher IMR values were noted among patients with CMD (CFR ≤ 2 and/or IMR ≥ 25) when compared to the group without CMD in both the high and low FFR groups separately (Appendix A).

### 3.7. Treatment Adjustment after Coronary Microcirculation Assessment

Every fifth patient (21.1%) underwent PCI. Patients with normal IMR were more likely to be revascularised than those with impaired IMR (29.8% vs. 11.9%, *p* = 0.003). We have not observed any differences in PCI rates regarding CFR and CMD (Table 3 and Appendix A). Although no significant differences were noted in the conservative treatment adjustment rates, patients with CFR impairment were more often provided with additional angiotensin-converting enzyme inhibitors (ACEI) (13.2% vs. 5.1%, *p* = 0.04). Those with IMR abnormality were administered both dihydropyridine calcium channel blockers (DHP CCBs) and non-DHP CCBs, in comparison to patients with normal IMR (34.5% vs. 6.6%, *p* < 0.001 and 3.6% vs. 0%, *p* = 0.04, respectively, Table 3). What is more, patients with CMD, regardless of treatment choice, had trimetazidine (23.2%) and dihydropyridine CCBs (26.4%) added to pharmacotherapy more frequently than those without CMD, who were treated conservatively (6.8%) and by revascularisation (4.2% with *p* = 0.002 and 0% with *p* = 0.0001, respectively, Appendix A).

### 3.8. Multivariable Analysis

No angina symptoms were significantly associated with either IMR or CFR impairment. However, it was revealed that intermediate stenosis in the left anterior descending (LAD) artery was connected with a 77.1% lower risk of impaired IMR, whereas intermediate stenosis within the Cx artery was related to an 85.4% decreased risk of impaired CFR. Moreover, hyperlipidaemia was noted as a factor associated with a significantly lower risk of abnormal IMR and CFR at the same time, while coronary non-invasive diagnostic testing was associated with lower risk of abnormal IMR. However, severe valvular disease and longer hospitalisations were linked with a higher risk of impaired IMR, CFR and IMR values, respectively (Appendix A).

## 4. Discussion

In our analysis, we found that the frequency of coronary microcirculatory assessments in Poland has steadily increased. Secondly, patients with either impaired IMR or CFR were generally less burdened with comorbidities. The third major finding of this study is that patients with normal IMR underwent revascularisation attempts more frequently. Fourthly, ACEI, CBBs and DHP CCBs with trimetazidine were added to pharmacotherapy more often after microcirculation testing among patients with low CFR, high IMR and CMD presence, respectively. Last of all, multivariable analysis revealed no association between angina symptoms and IMR or CFR impairment.

In general, the clinical characteristics of patients with CMD are poorly understood, as there is a great variety of underlying pathologies, the understanding of which is limited by the paucity of available scientific evidence. It is nonetheless well-recognised that CMD is often symptomatic and may even account for up to 65% of angina symptoms in patients with normal coronary angiography [20,24,25,26]. What is more, its presence might contribute to a worse improvement of angina symptoms in patients with chronic coronary syndromes [27].

In the conducted study, patients with either impaired CFR or IMR suffered from angina symptoms more frequently; however, such an association was not confirmed in the multivariable model. This may have been driven by the broad group selection, as our research included both patients with epicardial stenoses and with non-obstructive CAD. Nevertheless, in certain studies, it has been reported that CMD does not significantly affect hard outcomes in the case of patent epicardial arteries [28]. However, in the clinical trial conducted by Ford et al. [29], it was concluded that patients with non-obstructive CAD, for whom stratified medical therapy based on invasive coronary function testing was implemented, evinced improvement in terms of angina symptom management.

In our analysis, impaired coronary flow was characterised by different clinical characteristics, including angiographic images, as compared to abnormal IMR. Since CFR is not microvascular-specific and may be affected by resting haemodynamics, some discrepancies are anticipated to emerge between IMR and CFR values. In addition, low coronary flow may enlarge RFR/FFR discrepancy [30]. It was also reported that post-procedural FFR values and coronary flow significantly vary across the IMR value range [31]. Nevertheless, that does not undermine the value of combined measurements regarding microcirculation function. Instead, it indicates the complementary characteristics of these measurements.

Importantly, we observed that patients with normal IMR underwent PCI attempts more often that those with a high IMR. This reflected the fact that, in our study, patients with normal IMR had significantly lower median values of RFR and FFR than those with abnormal IMR. It is self-evident that the angiographic evaluation determines the PCI attempt. 

Interestingly, the majority of patients with abnormal IMR in our study were characterised by high FFR (>0.80), which has been a gold standard for functional epicardial coronary stenoses testing [32]. Lee et al. [7] reported that, in this group, patients with low CFR were more likely to experience patient-oriented composite outcomes (POCO; any death, MI, necessity of revascularisation) during the follow-up period, which was confirmed by other authors [33]. Furthermore, patients with the co-impairment of IMR and CFR had the highest POCO. In short, the association between microvascular impairment and a higher risk of cardiovascular events in patients with non-obstructive CAD has been confirmed in multiple studies [34,35,36,37].

In general, specific treatment options for patients with microcirculation abnormalities are limited and in contemporary clinical practice, they target symptom relief and mitigation of epicardial artery stenoses. However, in some studies, it has been suggested that patients with impaired CFR and IMR should receive more aggressive therapy [1]. We noted that, after microcirculation assessments, patients with low coronary flow or high IMR were more often provided with additional medications, i.e., ACEIs and CCBs. It could be anticipated that impaired CFR and IMR are linked to higher ACEI, CCB and statin therapies, as all of them are shown to decrease microvascular tone [26]. In fact, previously conducted clinical trials on microvascular angina were able to report improvements in coronary flow reserve following ACEI therapy as well as in angina symptom control after CCB drug implementation [38,39,40]. Moreover, we also noticed that the group with CMD (CFR ≤ 2 and/or IMR ≥ 25), regardless of treatment choice, had new DHP-CCBs and trimetazidine added to postprocedural pharmacotherapy more often than those without CMD treated conservatively or by PCI, with the latter receiving such medications least often. Trimetazidine, similarly to ranolazine, is usually given to patients as part of angina symptom management [41]. However, in a single blinded, randomised study conducted by Ilic et al. [42], it was shown that trimetazidine given to a group undergoing PCI also reduced microvascular dysfunction. This was expressed as a postprocedural IMR value. Nevertheless, it has to be emphasized that, due to the low FFR group involvement in the study and a lack of association between angina symptoms and CFR or IMR impairment, treatment choices may have been primarily driven by the presence or absence of obstructive atherosclerosis.

What is more, we noted that Cx and LAD artery location of intermediate stenosis, hyperlipidaemia and cardiac non-invasive diagnostic testing were associated with a decreased risk of microcirculation measurement abnormalities, whereas longer hospitalisation and severe valve disease increased the risk of impaired IMR and CFR. In other studies, it was reported that Killip class, delayed hospitalisation from symptom onset, peak troponin-I levels and multivessel disease were also linked to abnormal IMR [43]. Considering the risk factors, LDL-C, older age, female sex, hypertension, diabetes, previous MI and chronic renal failure were also shown to have associations with developing CMD and lower CFR [28,37,44].

In conclusion, although there is yet a great data paucity regarding targeted therapy for patients with CMD and their clinical characteristics, our analysis allows us to reveal contemporary treatment choices in clinical practice and certain associations between patients’ condition and microcirculation abnormality and, therefore, hypotheses are generated that require further investigation in prospective studies. 

### Limitations

This analysis, based on a national registry, has several limitations. Most importantly, it lacks a randomised design due to its retrospective characteristics. Hence, the results have to be considered solely as hypothesis-generating. Furthermore, due to the registry-based design, certain data were not available.

## 5. Conclusions

The frequency of coronary microcirculatory assessments in Poland has been steadily increasing. Angina symptoms were not associated with either IMR or CFR impairment. After microcirculation assessment, patients with impaired microcirculation, expressed as either low CFR, high IMR or both, received additional pharmacotherapy treatment more often. 

## Figures and Tables

**Figure 1 medicina-60-00277-f001:**
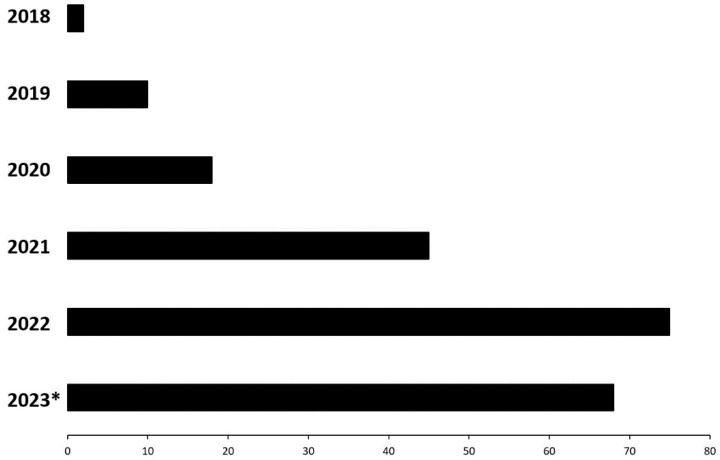
Coronary microcirculatory assessment cases in the last five years. * Data from June 2023.

**Figure 2 medicina-60-00277-f002:**
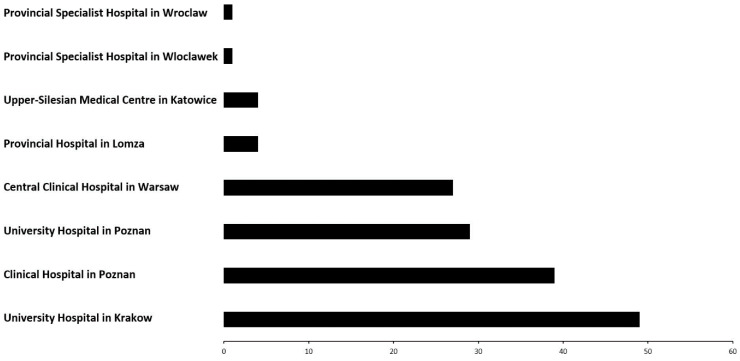
Coronary microcirculatory assessment number per CathLab registries in Poland.

**Table 1 medicina-60-00277-t001:** Clinical presentation.

Variable	Total N = 223	CFR ≤ 2 N = 91	CFR > 2 N = 117	*p*-Value	IMR ≥ 25 N = 84	IMR < 25 N = 121	*p*-Value
NYHA class							
None	105 (47.1)	41 (45.1)	51 (43.6)	0.79	36 (42.9)	54 (44.6)	0.19
I	29 (13.0)	10 (11.0)	19 (16.2)		7 (8.3)	22 (18.2)	
II	63 (28.3)	27 (29.7)	34 (29.1)		27 (32.1)	33 (27.3)	
III	21 (9.4)	11 (12.1)	10 (8.5)		12 (14.3)	9 (7.4)	
IV	5 (2.2)	2 (2.2)	3 (2.6)		2 (2.4)	3 (2.5)	
Shortness of breath	81 (37.5)	44 (48.4)	35 (29.9)	0.01	37 (44.0)	41 (33.9)	0.14
Chest pain	148 (68.5)	70 (76.9)	71 (61.7)	0.02	64 (77.1)	75 (62.5)	0.03
Palpitation/arrhythmia	34 (15.7)	16 (17.6)	17 (14.5)	0.55	19 (22.6)	13 (10.7)	0.02
Syncope	6 (2.8)	2 (2.2)	4 (3.4)	0.47	2 (2.4)	4 (3.3)	0.52
Prior cardiac arrest	5 (2.4)	1 (1.1)	4 (3.5)	0.27	2 (2.4)	3 (2.5)	0.67
CCS class							
None	61 (27.4)	24 (26.4)	30 (25.6)	0.20	15 (17.9)	38 (31.4)	0.03
I	30 (13.5)	8 (8.8)	21 (17.9)		8 (9.5)	21 (17.4)	
II	94 (42.2)	39 (42.9)	50 (42.7)		44 (52.4)	43 (35.5)	
III	37 (16.6)	19 (20.9)	16 (13.7)		17 (20.2)	18 (14.9)	
IV	1 (0.5)	1 (1.1)	0 (0.0)		0 (0.0)	1 (0.8)	
Stable angina symptomatic	100 (44.8)	43 (47.3)	52 (44.4)	0.69	44 (52.4)	50 (41.3)	0.12
Unstable angina symptomatic	36 (16.1)	20 (22.0)	14 (12.0)	0.052	17 (20.2)	67 (79.8)	0.24

Data are presented as counts (percentages). CCS: Canadian Cardiovascular Society; CFR: coronary flow reserve; IMR: index of microcirculatory resistance; NYHA: New York Heart Association.

**Table 2 medicina-60-00277-t002:** Angiographic findings.

Variable	Total N = 223	CFR ≤ 2 N = 91	CFR > 2 N = 117	*p*-Value	IMR ≥ 25 N = 84	IMR < 25 N = 121	*p*-Value
Angiographic image							
- no changes	32 (15.5)	11 (12.4)	20 (17.7)	0.30	12 (14.6)	19 (16.1)	0.78
- mild stenosis	166 (79.8)	74 (84.1)	88 (76.5)	0.18	71 (85.4)	89 (75.4)	0.08
- intermediate stenosis	127 (61.1)	53 (59.6)	70 (61.4)	0.79	34 (41.0)	87 (73.7)	<0.001
- significant stenosis	30 (13.5)	17 (18.7)	12 (10.3)	0.08	9 (10.7)	20 (16.5)	0.24
- CTO	12 (5.4)	5 (5.5)	7 (6.0)	0.88	1 (1.2)	11 (9.1)	0.02
Intermediate stenosis location							
- LMCA	7 (3.1)	4 (4.4)	3 (2.6)	0.70	25 (29.8)	7 (5.8)	0.04
- LAD	96 (43.0)	39 (42.9)	56 (47.9)	0.47	8 (9.5)	68 (56.2)	<0.001
- Cx	26 (11.7)	15 (16.5)	9 (7.7)	0.049	8 (9.5)	16 (13.2)	0.42
- RCA	16 (7.2)	8 (8.8)	8 (6.8)	0.60	6 (7.1)	8 (6.6)	0.45
- Mg	2 (0.9)	7 (7.7)	2 (1.7)	0.50	3 (3.6)	2 (1.7)	0.51
- LVB	0 (0.0)	0 (0.0)	0 (0.0)	-	0 (0.0)	0 (0.0)	-
- RPD	0 (0.0)	0 (0.0)	0 (0.0)	-	0 (0.0)	0 (0.0)	-
Significant stenosis location							
- LMCA	0 (0.0)	0 (0.0)	0 (0.0)	-	0 (0.0)	0 (0.0)	-
- LAD	12 (5.4)	3 (3.3)	4 (3.4)	0.17	1 (1.2)	5 (4.1)	0.35
- Cx	9 (4.0)	9 (9.9)	6 (5.1)	0.52	1 (1.2)	6 (5.0)	0.63
- RCA	12 (5.4)	1 (1.1)	3 (2.6)	0.03	0 (0.0)	11 (9.1)	0.02
- Mg	0 (0.0)	0 (0.0)	0 (0.0)	-	0 (0.0)	0 (0.0)	-
- LVB	0 (0.0)	0 (0.0)	0 (0.0)	-	0 (0.0)	0 (0.0)	-
- RPD	1 (0.4)	0 (0.0)	0 (0.0)	0.26	0 (0.0)	1 (0.8)	0.40
CTO location							
- LAD	4 (1.8)	1 (1.1)	3 (2.7)	0.63	0 (0.0)	4 (3.3)	0.15
- Cx	1 (0.4)	4 (4.4)	1 (0.9)	1.0	0 (0.0)	1 (0.8)	1.0
- RCA	8 (3.6)	0 (0.0)	4 (3.4)	0.73	0 (0.0)	7 (5.8)	0.15

Data are presented as counts (percentages). CFR: coronary flow reserve; CTO: chronic total occlusion; Cx: circumflex (artery); IMR: index of microcirculatory resistance; LAD: left anterior descending (artery); LMCA: left main coronary artery; LVB: left ventricular bypass; Mg: marginal (artery); RCA: right coronary artery; RPD: right posterior descending (artery).

**Table 3 medicina-60-00277-t003:** Treatment after coronary microcirculatory assessment.

Variable	Total N = 223	CFR ≤ 2 N = 91	CFR > 2 N = 117	*p*-Value	IMR ≥ 25 N = 84	IMR < 25 N = 121	*p*-Value
PCI	47 (21.1)	21 (23.1)	26 (22.2)	0.88	10 (11.9)	36 (29.8)	0.003
PCI within:							
LAD	36 (16.1)	16 (17.6)	20 (17.1)	0.93	8 (9.5)	27 (22.3)	0.02
LMCA	1 (0.5)	1 (1.1)	0 (0.0)	0.44	0 (0.0)	1 (0.8)	0.59
Cx	9 (4.0)	3 (3.3)	6 (5.1)	0.52	2 (2.4)	7 (5.8)	0.31
RCA	6 (2.7)	4 (4.4)	2 (1.7)	0.25	0 (0.0)	6 (5.0)	0.04
CABG	5 (2.2)	3 (3.3)	2 (1.7)	0.66	1 (1.2)	4 (3.3)	0.40
Percutaneous valve intervention	1 (0.5)	1 (1.1)	0 (0.0)	0.45	1 (1.2)	0 (0.0)	0.43
Surgical valve intervention	1 (0.5)	1 (1.1)	0 (0.0)	0.44	0 (0.0)	1 (0.8)	1.0
Conservative treatment	197 (88.3)	78 (85.7)	104 (88.9)	0.49	78 (92.6)	101 (83.5)	0.047
Conservative treatment adjustment	39 (17.5)	12 (13.2)	26 (22.2)	0.385	11 (13.1)	26 (21.5)	0.62
Added pharmacotherapy							
ACEI	18 (8.1)	12 (13.2)	6 (5.1)	0.04	10 (11.9)	8 (6.6)	0.19
BB	12 (5.4)	5 (5.5)	7 (6.0)	0.88	6 (7.1)	6 (5.0)	0.51
Nitrate	12 (5.4)	3 (3.3)	9 (7.7)	0.18	4 (4.8)	8 (6.6)	0.58
Ranolazine	0 (0.0)	0 (0.0)	0 (0.0)	-	0 (0.0)	0 (0.0)	-
Ivabradine	1 (0.4)	1 (1.1)	0 (0.0)	0.26	1 (1.2)	0 (0.0)	0.23
Trimetazidine	35 (15.7)	19 (20.9)	16 (13.7)	0.17	19 (22.6)	15 (12.4)	0.053
CCB DHP	38 (17.0)	20 (22.0)	17 (14.5)	0.16	29 (34.5)	8 (6.6)	<0.001
CCB NDHP	4 (1.8)	3 (3.3)	1 (0.9)	0.20	3 (3.6)	0 (0.0)	0.04

Data are presented as counts (percentages). Abbreviations: see Table 2. CABG: coronary artery bypass graft; DHP: dihydropyridine; PCI: percutaneous coronary intervention; NDHP: nondihydropyridine.

## Data Availability

Data are available upon a justified request.

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
