# Peer review of "Implementation of Microcirculation Examination in Clinical Practice—Insights from the Nationwide POL-MKW Registry"

_medicina, 2024, doi:10.3390/medicina60020277_

Round 1

Reviewer 1 Report

Comments and Suggestions for Authors

This is a well-reasoned and well-written research paper on Microcirculation Examination in Clinical Practice, which largely expands our understanding of the management of patients with abnormal microcirculation function.

 The main question addressed by the research: Patients with impaired microcirculation, expressed as either low CFR, high IMR or both, received additional pharmacotherapy treatment more often
it is relevant and interesting.
it is original topic.
There is limited data regarding the characteristics of patients who undergo microcirculation testing.
 it is well-written.The text is clear and easy to read.
the conclusions consistent  are with the evidence and arguments presented.  They address the main question posed.

Comments on the Quality of English Language

Moderate editing of English language required.

Author Response

Dear Dr. Januszek,

Thank you for your submission. Authors are encouraged to provide a

graphical representation of the paper as a self-explanatory image to

appear alongside with the abstract appearing on the Table of Contents.

It will help attract more readers and increase the visibility of your

paper once it is published online.

The GA should be a high-quality illustration or diagram in any of the

following formats: PNG, JPEG, TIFF, or SVG. Written text in a GA should

be clear and easy to read, using one of the following fonts:

Times, Arial, Courier, Helvetica, Ubuntu or Calibri. The minimum

required size for the GA is 560 × 1100 pixels (height × width), The size

should be of high quality in order to reproduce well. The GA should not

be totally same as a Figure in the manuscript. When submitting larger

images, please make sure to keep to the same ratio.

Please kindly provide a graphic abstract to present your research and

send it to us via email, we will help upload it to the editorial system.

The GA was sent by e-meil.

Dear Dr. Januszek, We have just sent you the review reports. In addition to revising thepaper based on the reviewers' comments, according to the requirementsofthe journal, the following points also need to be revised: 1. Enrich the contents: The length of the present version is a littleshorter than what we expected for article paper. In order to increasethe readability of the article and to have a deeper understanding of theresearch content for readers, we encourage authors to publish theirresults and experimental methodology in as much detail as possible sothat results can be reproduced. We noticed that the main text of yourmanuscript is quite brief which may mean that the materials and methods,research background, future research directions, or possibleapplications of the research are not described in enough detail. This has been enriched.   2.Attached is the Authorship Contribution Form. Couldyou please provide us with some more details regarding each authorsindividual contribution?. We would also appreciate it if you could fillin and sign the attached Authorship Contribution Form. 

This has been attached.

Reviewer 1

This is a well-reasoned and well-written research paper on Microcirculation Examination in Clinical Practice, which largely expands our understanding of the management of patients with abnormal microcirculation function.

 The main question addressed by the research: Patients with impaired microcirculation, expressed as either low CFR, high IMR or both, received additional pharmacotherapy treatment more often
it is relevant and interesting.
it is original topic.
There is limited data regarding the characteristics of patients who undergo microcirculation testing.
 it is well-written.The text is clear and easy to read.
the conclusions consistent  are with the evidence and arguments presented.  They address the main question posed.

Moderate editing of English language required.

Thank you. English was corrected once more by native speaker.

Reviewer 2

The pertinence is related to the paucity of published data on the characteristics of patients undergoing microvascular testing, the comparison of clinical characteristics and subsequent decision-making after coronary microvascular assessment between patients with abnormal microvascular function and those with normal IMR and CFR values, and the assessment of factors associated with impaired microvascular measurements based on a national registry of invasive coronary microvascular testing (POL-MKW). The study is well designed and reported. The methods are fully described and the results are detailed and clear. The study is within the scope of the journal and is of potential interest to cardiovascular disease specialists.

Than you.

Reviewer 2 Report

Comments and Suggestions for Authors

The pertinence is related to the paucity of published data on the characteristics of patients undergoing microvascular testing, the comparison of clinical characteristics and subsequent decision-making after coronary microvascular assessment between patients with abnormal microvascular function and those with normal IMR and CFR values, and the assessment of factors associated with impaired microvascular measurements based on a national registry of invasive coronary microvascular testing (POL-MKW). The study is well designed and reported. The methods are fully described and the results are detailed and clear. The study is within the scope of the journal and is of potential interest to cardiovascular disease specialists.

Author Response

(The authors gave the same response as above.)
